# Obesogenic Lifestyle and Its Influence on Adiposity in Children and Adolescents, Evidence from Mexico

**DOI:** 10.3390/nu12030819

**Published:** 2020-03-19

**Authors:** Desiree Lopez-Gonzalez, Armando Partida-Gaytán, Jonathan C. Wells, Pamela Reyes-Delpech, Fatima Avila-Rosano, Marcela Ortiz-Obregon, Frida Gomez-Mendoza, Laura Diaz-Escobar, Patricia Clark

**Affiliations:** 1Clinical Epidemiology Research Unit, Hospital Infantil de México Federico Gómez, CP 06720 Mexico City, Mexico; dradesireelopez@gmail.com (D.L.-G.); panchitaflor@hotmail.com (P.R.-D.); monserratavilarosano@gmail.com (F.A.-R.); nutriologamarceortiz@gmail.com (M.O.-O.); fridag-mendoza@hotmail.com (F.G.-M.); dralmed@hotmail.com (L.D.-E.); 2Universidad Nacional Autonoma de México, CP 04510 Mexico City, Mexico; dr.partida.g@gmail.com; 3Childhood Nutrition Research Centre Population, Policy and Practice, UCL Great Ormond Street Institute of Child Health, London WC1N EH, UK; jonathan.wells@ucl.ac.uk

**Keywords:** children 1, adolescents 2, obesity 3, adiposity 4, lifestyle, body composition

## Abstract

Overweight (OW) and obesity (OB) during childhood/adolescence are major public health problems in Mexico. Several obesogenic lifestyle (OL) risk factors have been identified, but the burden and consequences of them in Mexican children/adolescents remain unclear. The objective of this study was to estimate the prevalence of OL components and describe their relationships with adiposity, and OW/OB. A population-based cross-sectional study of Mexican children/adolescents with nutritional assessment, data collection on daily habits and adiposity as fat-mass index (FMI) by dual-energy X-ray absorptiometry was performed. Individual OL-components: “inactivity,” “excessive screen time,” “insufficient sleep,” “unhealthy-diet”, were defined according to non-adherence to previously published healthy recommendations. *Results*: 1449 subjects were assessed between March 2015 to April 2018. Sixteen percent of subjects had all four OL-components, 40% had three, 35% had two, 9% had one, and 0.5% had none. A cumulative OL score showed a significant dose–response effect with FMI. The combination of inactivity, excessive screen time, and insufficient sleep showed the highest risk association to OW/OB and higher values of FMI. *Conclusions*: The prevalence of OL-components was extremely high and associated with increased adiposity and OW/OB. Several interventions are needed to revert this major public health threat.

## 1. Introduction

Considerable increases in the prevalence of overweight (OW) and obesity (OB) in children and adolescents have occurred in many developed and emergent countries. Mexico is no exception, as 33.2% and 36.3% of children and adolescents are affected by OW/OB [1].

Childhood and adolescence are crucial periods for the formation of habits relevant to nutritional status and health. Moreover, lifestyle behaviors that are developed during this period have the potential to affect future adult health [2]. Several lifestyle habits have been reported to contribute to the worldwide health problem of OW/OB, including: inactivity, insufficient sleep, unhealthy diet, excessive time spent watching video-games and TV rather than engaging in conventional recreational activities, and increased availability of, and growing preferences for, high-energy density foods. Such a combination of behaviors reflects exposure to an obesogenic lifestyle (OL) [3]. To address this problem, specific healthy recommendations on diet and daily habits have been published by several organizations [4,5,6,7,8]. Lack of knowledge or non-adherence to such recommendations can result in an increased risk of adverse health-related outcomes.

The different OL habits generate a sustained positive energy balance, where energy intake is excessive relative to expenditure. Specific diet patterns together with increased sedentarism have been associated with adipose tissue accumulation and an increased risk of associated comorbidities [9,10]. Conversely, regular physical activity and a high intake of fruits, vegetables, whole-grain foods, milk, and fish are associated with an adequate glycemic index diet and a lower risk of adipose tissue accumulation and resulting comorbidities [4,5,6,11,12].

Because of its simplicity, BMI has become the standard tool to diagnose OW/OB, and it has shown a good correlation with adiposity in epidemiological studies. Nevertheless, at the individual level, BMI may misclassify subjects at risk [13,14] especially in populations were reference values are not reported, as in the case of Mexican children and adolescents [13]. Having a more direct measurement of adiposity may provide more reliable data to identify relationships between OL habits and their impact on health.

To understand the contribution of the OL to the obesity epidemic in Mexican children and adolescents, the aim of this study was to estimate the prevalence of specific OL components among children and adolescents aged 6–18 years old from Mexico City and test their associations with adiposity and OW/OB.

## 2. Materials and Methods

### 2.1. Study Participants

In a population-based cross-sectional study, children (6–11 years old) and adolescents (12–17 years old) from Mexico City and the metropolitan area were the target population. In accordance with the National Institute of Statistics and Geography (INEGI), in 2015, there were 1,319,293 children 5 to 14 years old, and 97% of them were attending school [15].

A polyetapic random sampling from 7511 registered schools at the *Secretaria de Educación Pública* (SEP) was selected. For stratification, counties, primary, middle or high school level, and public or private status of schools were variables taken into account. Finally, 15 schools where eligible and 13 agreed to participate. Written invitations were delivered to parents. Interested subjects showed their willingness to participate by scheduling an appointment via telephone between March 2015 and April 2019. The family and friends of recruited subjects that met the inclusion criteria were also invited. A total of 2155 appointments were scheduled, and 1767 subjects interested in the study attended their appointment and were evaluated for the following inclusion criteria: healthy individuals, without known chronic, endocrine, systemic, respiratory, neurological, cardiac, or psychiatric disorders; individuals without chromosomal diseases, genopathies, dysmorphic syndromes; and individuals who were not receiving pharmacological treatment affecting their lipid or glucose metabolism. Finally, 1463 subjects met the inclusion criteria, 12 were excluded for alterations in the pediatric examination, and two for missing values in the measurements. All subjects were recruited voluntarily.

This study was carried out according to the Helsinki Declaration of Clinical Research on Humans [16], and was reviewed, approved and registered by our Institutional Research, Ethics and Biosafety Committees (HIM2015-055). Parents or legal guardians who agreed to participate were asked to provide written informed consent, and children over 7 years gave their consent to participate.

### 2.2. Measurements

Subjects who agreed to participate in the study attended a single study visit where a full pediatric and nutritional assessment was carried out, including pubertal development according to the Tanner & Whitehouse scale [17,18,19]. Weight and height were measured with subjects wearing lightweight clothing, using a SECA® 284 scale stadimeter. Body mass index (BMI) was calculated as weight (kg) divided by the square of height (m) [20].

Adiposity was measured by dual-energy X-ray absorptiometry (Lunar-iDXA by GE Healthcare®) according to the manufacturer’s instructions and analyzed through Encore® software version 15 obtaining total body fat (g); truncal fat (g), appendicular fat (g), android fat (g), gynoid fat (g), and fat percentage relative to total body weight (%). Fat mass index (FMI) was calculated as total body fat (kg) divided by the square of height (m). We used FMI as the main adiposity measurement, because this index incorporates adjustment for height.

For BMI classification, we used conventional centile cut-off values from the growth charts of the Center for Disease Control and Prevention (CDC) to classify subjects as: underweight (<5th percentile), healthy weight (5th to < 85th percentile), overweight (85th to < 95th percentile) and obese (≥95th percentile) [21].

### 2.3. Habits Assessment and OL Components Definitions

Previous healthy recommendations have been published by several organizations. For the purpose of this study, the descriptions of the following are relevant:The World Health Organization (WHO) recommends ≥60min/day of moderate to vigorous physical activity [5];The Canadian Pediatric Society (CPS) recommends a maximum daily time dedicated to screens of 2 h [12];The American Academy of Sleep Medicine (AASM), the Sleep Foundation (SF), and the American Academy of Pediatrics (AAP) recommend minimum sleep duration of 9hr/night for children and 8hr/night for adolescents [6,7,22];The American Heart Association (AHA) recommends consumption of vegetables and fruits as ≥ 4.5 cups per day; fish 3.5-oz servings ≥ 2 times per-week; sodium, ≤ 1,500 mg per day; sugar-sweetened beverages, ≤ 450 kcal (36 oz) per week; and whole grains, ≥ 3 servings per day scaled to a 2,000-kcal/ per-day diet [4]. In fact, the AHA has proposed the Diet Score as ideal, intermediate, or poor, according to adherence to 4–5, 2–3 or 0–1 recommendation respectively [4]. To assess the diet score, we adapted the following items for pediatric population servings: vegetables and fruits ≥4.5 age-specific portions per day; fish ≥2 age-specific portions per week; sodium, ≤1500 mg per day; sugar-sweetened beverages, ≤4.5 portions (240 ml = portion) per week; and whole grains as total fiber 14 g/1000 kcal. Age-specific portions are adjusted for recommended age-specific daily calories intake recommendations, according to AAP [23] We assessed diet quality according to the AHA Diet Score because we find these recommendations reliable, simple to do, easy to teach to families, and allow for comparisons of the different components of the diet.

We collected data on daily habits such as sleep hours during weekdays and weekends (i.e., time to go to sleep at night and time to awake on a school-day and on a weekend day); daily time dedicated to watching television, doing homework or playing with the computer, tablet, cell phone or videogames (i.e., on a school day and on a weekend day); daily time dedicated to structured physical activity; and other kinds of physical activity like walking or playing. These data were obtained by structured interview with direct questions and by the self-administered International Physical Activity Questionnaire (IPAQ) in its Spanish version [24] under the parent’s or legal guardian’s supervision.

In order to assess the AHA Healthy Diet Score, diet habits were assessed by two questionnaires: a 24-hour-recall survey and a local validated 12-month food frequency questionnaire (FFQ). Briefly the FFQ includes 133 food items in the following categories: dairy products, carbohydrates, fats, proteins, vegetables, fruits, water, beverages with and without added sugar, and junk food (sweetened beverages and junk food was adapted for the pediatric population) [25]. The FFQ was used to estimate the weekly consumption of fish, sweetened beverages, and the daily consumption of fruits and vegetables. The FFQ has been previously validated and used to assess and inform diet habits in the Mexican population [25], though it is not the best source for more specific data (i.e., sodium and fiber daily consumption). Data from the 24-h-recall survey was specifically used to complement the FFQ by estimating the daily consumption of sodium and fiber. The 24-hour survey allowed for specific questions and more specific assessments (i.e., Was food fresh or processed? Did it have condiments, flavor bouillon, or other food seasonings?). These questionnaires were applied by experienced nutritionists and included the assessment of portion sizes [23]. Energy and macronutrients’ intake data were analyzed in The Food Processor Nutrition Analysis Software® version 11.1.

Energy intake (EI) misreporting bias was handled by applying the Goldberg cut-off method adapted for children by Black [26,27], based on implausible values for the ratio of reported EI_rep_ to estimated basal metabolic rate (BMR_est_) and classifying data as under-reporting, over-reporting or plausible reports. We estimated BMR_est_ using Schofield’s formula with height and weight [28]. Estimated energy expenditure (EE_est_) was calculated by multiplying BMR_est_ x physical activity level (PAL).

Four different components of the OL were defined as follows: *a.* *Inactivity*: a child or adolescent who performs <60min/day of moderate to vigorous physical activity;*b.* *Excessive screen time*: a child or adolescent who stays sitting or lying >2hr/day for screen time;*c.* *Insufficient sleep*: parents reporting sleep duration < 9hr/night for children and <8 hr/night for adolescents;*d.* *Unhealthy diet*: a children or adolescent who scored below 4 on the AHA Healthy diet score (i.e., intermediate or poor diet became a single variable: “unhealthy diet”).

### 2.4. Statistical Analysis

Conventional descriptive statistics were used to report the demographics and general characteristics of the sample, including the prevalence of individual OL components. Values are expressed as means ± standard deviations of the mean or absolute numbers and percentages according to the nature of variables. A comparison of the general characteristics between sexes in each age group was made by student’s *t-*test and chi-square test according the type of variable.

Mean differences of FMI stratified by age and sex groups were analyzed by unpaired *t*-tests. Proportion differences for BMI-groups stratified by age and sex groups were assessed by chi-squared tests. Pearson’s correlation coefficient was used to assess the relationship between estimated EE_est_ and reported EI_rep_.

The mean differences of FMI, stratified by age and sex groups and according the presence or absence of each OL component, were assessed by unpaired *t*-test. For the AHA Diet score, the mean differences assessment was done by ANOVA. Independent risk associations stratified by the age group and sex of each OL component for OW/OB are expressed as odds-ratio (OR), and their corresponding 95% confidence interval (95%CI) by a simple logistic regression model.

The mean differences of FMI, with respect to having 0, one, two, three or all four OL components stratified by age and sex groups, were done by ANOVA, and adjusted for multiplicity (Bonferroni *P* < 0.05) in order to find which is the different group. A multiple linear regression was done to estimate the effect size of the accumulation of OL components to FMI stratified by age and sex groups. A multiple linear regression analysis stratified for sex and age group and adjusted for Tanner stage was performed to test the independent effect of each OL component in FMI. Risk associations stratified by age and sex groups between the cumulative score of OL components to OW/OB are expressed as OR and their 95%CI by logistic regression model.

Two-step cluster analyses, taking OL components as independent categorical multinomial variables and the number of clusters automatically determined based on log-likelihood distances, were done. The resulting clusters were used to explore the specific combinations of OL components that may discriminate different risk associations to OW/OB and belongingness to the top-quartile, FMI values and expressed as OR and their 95%CI.

All available data were analyzed using SPSS IBM® software (version 20.0), without replacement of missing values. Values of *P* < 0.05 were considered to be statistically significant where applicable.

## 3. Results

A total of 800 children aged from 6 to 11.9 years (349 girls and 451 boys) and 649 adolescents aged from 12 to 17.9 years (309 female and 340 male) were included in the present analysis. Demographic, anthropometric, Tanner stage and general data of lifestyle habits are shown in Table 1.

### 3.1. OL Components

OL components were highly prevalent in the sample. Summary data on physical activity, screen time, sleep duration, AHA Diet score and intakes of energy and macronutrients are shown in Table 1.

Regarding physical activity, 58% of females and 80% of males reported performing any type of physical activity ≥ 1 occasion per week; included in these figures were 48% of females and 65% of males who reported performing structured physical activity on a regular basis with a mean duration of 3.0 h/w (S.D. 2.7 h/w). According to the WHO recommendations on physical activity, 84% of the subjects were *inactive*. Specifically, 92% (*n* = 322) of female children, 86% (*n* = 387) of male children, 91% (*n* = 277) of female adolescents, and 75% (*n* = 256) of male adolescents, were inactive.

The assessments of screen time showed that 46% (*n* = 161) of female children, 49% (*n* = 223) of male children; 73% (223) of female adolescents and 69% (*n* = 235) of male adolescents exceeded the recommendations of the CPS. The mean screen time was 3 h/d (S.D. 1 h/d) for children and 4 h/d (S.D. 2 h/d) for adolescents by both sexes (see Table 1).

Of the four OL components, *insufficient sleep* was the least prevalent in the sample, reported by 31% (*n* = 108) of female children, 25% (*n* = 113) of male children, 42% (*n* = 129) of female adolescents and 36% (*n* = 122) of male adolescents. The means of sleep duration were 9 h/d (S.D. 1 h/d) for children, and 8 h/d (S.D. 1 h/d) for adolescents in both sexes (see Table 1).

In the energy intake evaluation, in total, 223 subjects (15%) of the sample were classified as over-reporting, 43 (3%) as under-reporting and 81% as plausible reports. Goldberg cut-off values are shown in Appendix A. Misreporting frequency of the sample according to BMI-categories is shown in Appendix A. Reported EI correlated moderately with calculated EE (Pearson’s r = 0.283; P ≤ 0.01). All analyses were done with the total sample (*n* = 1449) and with the plausible-reports subsample (*n* = 1171), with no significant differences between both. Following the parsimony principle, results are only shown for the total sample.

A healthy diet was reported only by 11 (3%) female children, 11 (2%) male children, nine (3%) female adolescents and eight (2%) male adolescents. More than 50% of the sample scored as poor diet according to the AHA Diet Score. One hundred and eighty-eight (54%) female children, 247 (55%) male children, 247 (55%) female adolescents and 179 (53%) of male adolescents reported a poor diet score. Noticeably, 92% of the sample exceeded the recommended sodium intakes, followed by 74% exceeding the recommended ingestion of sweetened beverages. Most subjects (55% to 67% of the total sample) did not eat enough fruit and vegetables, fish, or fiber. 

We observed inverse and significant correlations between the adequate consumption of fiber with adiposity (FMI) (beta coefficients -0.66 kg/m^2^ (95%CI -1.2 to −0.1; *P* = 0.02) for female children, −0.85 kg/m^2^ (95%CI −1.4 to −0.3; *P* = 0.002) for male children, −0.9 kg/m^2^ (95%CI −1.6 to −0.2; *P* = 0.012) for female adolescents, and −0.92 kg/m^2^ (95%CI −1.7 to −0.2; *P* = 0.015) for male adolescents).

Regarding the accumulation of OL components, 44 (13%) female children, and 48 (11%) male children reported having all four OL components, 136 (39%) and 163 (36%) reported three, 134 (38%) and 185 (41%) reported two, and 35 (10%) and 51 (11%) reported one OL component, respectively. No female children and only four (1%) male children reported no OL components.

For the adolescent groups, 74 (24%) females and 61 (18%) males reported all four OL components; 141 (46%) and 136 (40%) reported three, 76 (25%) and 105 (31%) reported two, 14 (5%) and 36 (11%) reported one OL component and only one female (0.3%) and two males (1%) reported no OL components.

### 3.2. Measurements of Adiposity

According to BMI categorization, the prevalence of OW in children was 14.6% (95% CI 12.2, 17.3), for female children 15.8% (95% CI 12.3, 19.95), male children 13.7% (95% CI 10.88, 17.24); and 18% (95% CI% 15.1, 21.2) in adolescents, for female adolescents was 23.6% (95% CI 19.23, 28.66) and male adolescents 12.9% (95% CI 9.78, 16.93.3). The prevalence of obesity was 18% (95% CI 15.4, 20.8) in children, for female children 15.8% (95%CI 12.3, 19.95), male children 19.7% (95% CI 16.32, 23.65) and 14% (95% CI 11.4, 16.9) in adolescents, for female adolescents the prevalence was 13.3% (95% CI 9.93, 17.51) and for male adolescents the prevalence was 14.7% (95%CI 11.34, 18.17). The different measurements of adiposity are shown in Table 2, stratified by age and sex groups. Female adolescents showed significantly more adiposity in all measurements; female children only showed increased gynoid fat, fat percentage, and FMI when compared to male children.

Linear regression analyses showed a significant positive relation between FMI and BMI, increasing 3.4 kg/m^2^ (95% CI 3.2, 3.6, *P* < 0.001) for each ascending BMI category. Age and sex groups modified this relationship, with 2.9 kg/m^2^ (95% CI 2.7, 3.2, *P* < 0.001) in female children to 3.8 kg/m^2^ (95% CI 3.4, 4.1, *P* < 0.001) in female adolescents, and 3.3 kg/m^2^ (95%CI 3.1, 3.5, *P* < 0.001) for male children, and 3.6 kg/m^2^ (95% CI 3.3, 3.9, *P* < 0.001) for male adolescents. In all the datasets, the correlation between FMI and BMI was good (r = 0.89; *P* < 0.001).

The misclassification of adiposity using BMI against adiposity measured by DXA was found in 5.2% of the sample. The distribution of FMI across the different BMI categories showed that up to 4.5% of female children, 5.1% of male children, 4.7% of female adolescents and 1.8% of male adolescents classified as “healthy” weight according to BMI percentile had FMI mean values equal to or greater than the mean of their OW peers (i.e., FMI 7.44 kg/m2 for OW female children; 6.87 kg/m2 for OW male children, 10.13 kg/m2 for OW female adolescents, and 7.52 kg/m2 for OW male adolescents). This was also true for 0.9% of female children, 0.7% male children, 0.5% female adolescents and 0.9% of male adolescents classified as a “healthy” weight by BMI, who likewise showed FMI mean values equal to or greater than the mean of obese peers (10.53 kg/m2 for OB female children; 10.13 kg/m2 for OB male children; 13.45 kg/m2 for OB female adolescents, and 10.68 kg/m2 for OB male adolescents).

The means of FMI according to the presence or absence of individual OL components are shown in Table 3. Significant differences were observed in relation to *inactivity* in female children and male adolescents, *excessive screen time* in both female and male children, and *insufficient sleep* only in male children.

Since there are no published cut-off point values for FMI, we undertook univariate analyses to evaluate the risk associations of each OL component to OW/OB, stratified by age and sex groups. We found that only *inactivity* was independently associated with OW/OB, showing an OR of 6.55 (95% CI 1.52, 28.22; *P* = 0.004) in female children, and in male adolescents an OR of 2.2 (95% CI 1.2, 4.1; *P* = 0.012). Table 4.

A significant dose–response relationship between the cumulative numbers of OL components (OL cumulative score) and FMI was evident for children and is shown in Table 5. Male children without OL components, showed an FMI mean of 3.6 kg/m^2^ compared to 5.2, 5.7, 5.9, and 6.9 kg/m^2^ among those with one, two, three, or all four OL components, respectively. Regarding female children, no subject had zero OL components; the mean FMI in the group with one OL component was 5.2 kg/m^2^ compared to 6.0, 6.5, and 7.0 kg/m^2^ in those with two, three, or four OL components. The generalized linear model revealed that, for each OL component, the FMI increased by 0.539 kg/m^2^.

The same analyses in adolescents showed a tendency for FMI to increase in association with the cumulative score of OL components, but this effect was not significant (see Table 5).

The relationship between each OL component and FMI, assessed by multiple regression analysis stratified by age group, sex and adjusted for Tanner, is shown in Table 6. Relevant associations were only observed for inactivity in female children, excessive screen time for male children, insufficient sleep for male children and Tanner stage for all groups except male children.

The OL cumulative score also showed a significant positive risk association for OW/OB. We set as reference group those subjects with 0 or one OL component and compared them to subjects having two, three, or four OL components, by a risk association stratified by sex and age groups and shown in Table 7.

Two-step cluster analyses revealed different groups according to specific combinations of OL components. For the whole sample, we found the highest risk association for OW/OB in those subjects who reported *inactivity*, *excessive screen time*, and *insufficient sleep* with an OR of 4.7 (95%CI 2, 11.6; *P* = 0.0006) for OW/OB and an OR of 5.5 (95%CI 1.9, 15.8; *P* < 0.001) for belongingness to the top quartile of FMI. When stratified by age and sex groups, we observed different effects: female children showed an OR of 1.5 (95% CI 1.22, 1.88; *P* = 0.039) for OW/OB, and an OR of 1.48 (95% CI 1.21, 1.81; *P* = 0.045) for belongingness to top quartile FMI values; for male children OR = 8.8 (95% CI 1.87, 41.6; *P* = 0.006) for OW/OB and OR = 13.9 (95% CI 1.7, 111.3; *P* = 0.002) for top quartile FMI values. Significance was lost for both sexes in the adolescent group (i.e., female adolescents OR = 1.12 (95%CI 0.19, 6.5; *P* = 0.8935) for OW/OB and OR = 1.7 (95% CI 0.19, 15.5; *P* = 0.62) for top quartile FMI values; male adolescents OR = 4.36 (95% CI 0.92, 20.65; *P* = 0.063) and OR = 3.03 (95% CI 0.64, 14.4; *P* = 0.14)).

## 4. Discussion

Overall, this study has estimated a very high prevalence of OL components in urban Mexican children and adolescents, and a significant risk association with increased adiposity, overweight and obesity. To the best of our knowledge, this is the first study of Mexican children and adolescents that evaluates the effect of OL components on adiposity, measured as fat mass by DXA.

The sample size of the study was large and evenly distributed by sex as well as by the different age groups. Our sample shows a similar distribution of nutritional status according to BMI as the National Study reported in 2016 by the National Health and Nutrition survey (ENSANUT) [1], providing indirect evidence that our sample is representative.

Previous studies have published the prevalence of OL components in children in different countries. The International Study of Childhood Obesity Lifestyle and Environment (ISCOLE) study was a cross-sectional, multinational study designed to determine the relationship between lifestyle behaviors and obesity in 12 countries [29]. The Identification and Prevention of Dietary and Lifestyle-induced Health Effects in Children and Infants (IDEFICS) study reported the etiology of childhood obesity in eight countries [30]. We chose these two studies to compare our observations based on the prevalence of these OL components.

Regarding physical activity, IDEFICS reported 21% of their normal weight/thin and 12% of OW children meeting the WHO recommendation [30], and ISCOLE reported 44% of their participants meeting the WHO recommendation [29]. Mexican children and adolescents in our study reported significantly less daily moderate to vigorous physical activity (MVPA), with only 11% of children and 18% of adolescents meeting the WHO recommendation. Concerning time dedicated to screen, 48% of Mexican children and 71% of Mexican adolescents showed excessive screen time in this study. This OL component was lower prevalent in children than data reported by ISCOLE (3.2 h ± 1.8 vs. 2.9 h ± 1.8; *P* < 0.0001) [29] and IDEFICS (63.3% of normal/thin children and 56.5% of overweight/obese children spending < 2 h/day) [30]. The least frequent OL component in our sample was insufficient sleep; 28% of children and 39% of adolescents did not comply with the AASM, SF, AAP recommendations. When compared to ISCOLE (58% of their sample not meeting such recommendations) [29] and IDEFICS (58% of their normal weight/thin and 62% of OW children not meeting such recommendations) [30] Mexican children and adolescents seem to have better sleep, at least in terms of duration.

Interestingly, ISCOLE reported 7.2% of their subjects performing sufficient MVAP, spending < 2 h/day of ST and sleeping sufficiently, and this specific “healthy” behavior was associated with significantly less risk of obesity (OR 0.28, 95%CI 0.18, 0.45, *P* < 0.05) [29]. Aligned with this observation, we observed only 3.7% (95% CI 2.8, 4.8) of subjects in our sample with this healthy behavior, and it was also associated with a protective effect (OR of 0.24, 95%CI 0.10, 0.57; *P* = 0.001) for OW/OB and for belongingness to the top quartile FMI values (OR 0.24 95% CI 0.08, 0.66, *p* = 0.0057).

Concerning dietary habits, our study showed that >50% of Mexican children and adolescents reported a poor diet, including > 70% exceeding the recommended intake of sweetened beverages. These results are similar to the IDEFICS results, where 78.2% of their sample exceeded this recommendation [30]. More than 65% of our sample did not consume enough fiber or fish, and > 50% did not eat enough fruits or vegetables. According to Perng, Mexican diets are characterized by a high consumption of sweetened beverages, fried foods, and processed foods, and a low consumption of fresh fruits and vegetables [11], which was borne out in this study. Such a diet has been previously associated with increased adiposity [11].

Taken together, these observations provide supportive evidence of a very high prevalence of unhealthy lifestyle habits in children and adolescents around the world. The interrelationship between them may show a cumulative negative effect when present, but also a window for possible intervention.

In addition to describing the prevalence of these OL components in Mexican children and adolescents, we were interested to know their associations with OW/OB, and especially with adiposity measured by DXA.

Previous studies have also assessed the relationship between comparable lifestyle habits and adiposity, as measured by DXA. For instance, Kwon et al. published their observations assessing physical activity with accelerometers at age 5, 8, 11, 13, 15, 17 and 19 years old, and reported that subjects with a PA trajectory starting as active (> 45min/day of MVPA), but decreasing activity with age, showed a significantly higher risk of becoming obese, and an increased body fat percent when compared to subjects with a consistent active trajectory through the years, and even when they were compared to inactive subjects through the years [31]. In contrast, Hjorth et al. conducted a longitudinal study and reported that in children aged 8-11 years old, adiposity at baseline was a better predictor of lower physical activity and higher sedentary time after 200 days and not vice-versa [32]. Regarding these observations, we hypothesize that age, reported behavior at time of assessment and consistency through time may be very important factors to consider. These cannot be proven in our study due to the cross-sectional design, however, if all these observations are taken, then it is reasonable to think that, in our group, low physical activity and fat mass accumulation must be linked.

The specific barriers to doing more PA for Mexican children and adolescents are several and complex: time devoted to PA in private and public schools is limited to one hour per week. The scarcity of appropriate public places to do sports, coupled with the complexity of commuting and insecurity, are well known problems in urban Mexico. Adequate promotion and infrastructure for PA in our country need to be addressed by the authorities to help change this behavior and stimulate PA [33].

Other relevant study is that by Henderson et al., who explored the relationship between PA, fitness and sedentary behavior with insulin sensitivity (IS) in a high-risk population of Canadian children. They observed independent associations between physical activity and sedentary behavior (both measured by accelerometer) with IS, but these effects were lost when controlled for adiposity. The sex-specific differences found regarding MVPA, which were evident even after controlling for sedentary behavior, fitness and adiposity, are noteworthy [34]. In our sample, we observed a significant difference in physical activity between sex groups: female children dedicate less time to physical activity compared with male peers; this difference significantly increased between adolescents. These differences may be partly explained through the “Latin stereotype,” in which females are thought to participate in activities related to home and motherhood (they need to be caring and serve others), and not perform activities reserved for males, like sports (boys need to be competitive, aggressive, and strong). This attitude persists significantly in Mexican society as a whole [35]. Relevant to this sex-related difference in physical activity and its potential effects further in life is the acknowledgement of a known paradox: women show less cardiovascular risk when compared to male peers, and this has been related to estrogen, but, as they grow older and estrogen influence decreases, either naturally or driven by other factors, their cardiovascular risk equals their male peers [36,37,38]. Thus, the Latin stereotype and physical activity sex-related differences can have an adverse impact on the future cardiovascular risk of Mexican female children and adolescents.

Excessive time spent in front of a television or computer screen is a dominant sedentary pastime of all ages and is closely related to reduced physical activity, increased meal frequency, snacking, energy intake and adiposity [9,12,39]. In addition, many of the food products shown on TV are directed to children and are high in sugar and fat [40]. Problems with screen time are more complex than only sedentary behavior, and may be related to adverse effects on sleep, social interaction, and family life [41]. The new recommendations of the AAP suggest finding an equilibrium point between physical activity, healthy nutrition, good sleep hygiene and nurturing social environment [42]. In our sample, excessive screen time was significantly associated with higher adiposity in female and male children, although the Tanner stage attenuated the effect in female children.

Several studies have explored the relationship between sleep duration and/or quality with adiposity. The FLAME study reported that a short sleep duration between age 3 and 5 years old predicted higher FMI at age 7 years old [43]. We were only able to observe higher adiposity values in male children who reported insufficient sleep. Relevant to this discussion is the evidence provide by Collings et al., who reported that sleep duration in early childhood was inversely and independently associated with fat percent, measured by DXA 24 months later. Interestingly, when they analyzed the opposite relationship between adiposity in early childhood and sleep duration 24 months later, they also observed an inverse and independent association. Several plausible mechanisms have been proposed for the effect of sleep on adiposity, including appetite dysregulation via ghrelin/leptin, sleepiness-associated reduction in physical activity, and a preference for sedentary activities [44,45,46].

Regarding dietary habits, Gingras et al. explored associations between central adiposity measured with DXA and healthy dietary habits such as: a) eating breakfast daily, b) having daily family dinner, c) eating fast foods < 1/week and d) eating meals while watching television < 1/week. They observed that healthy dietary behaviors throughout childhood were associated with less adiposity and lower insulin resistance in adolescence. The longitudinal design of their study allows for a potential causal relationship between unhealthy lifestyle habits with the accumulation of adipose tissue and related adverse health outcomes [47].

Pereira-Rocha et al. reported the association of five dietary patterns found in their sample of Brazilian children aged 8-9 years old with adiposity as measured by DXA. In their results, sedentary children (i.e., watching television or engaging in other forms of screen-based entertainment ≥ 2h/day) with unhealthy diets had a higher and significant adiposity compared with children with healthy patterns [48].

Taken altogether, our results show a clear panorama of the most common plausible responsible mechanisms of this huge health problem in our population. Mechanistically, the high prevalence at early ages of these OL components expose children and adolescents to an environment that results in an unhealthy imbalance between energy intake and expenditure, with the potential to imprint their behaviors and transcend across generations. Aligned with these, several observations have been reported in countries that participated in the ISCOLE and IDEFICS studies and, in general, are consistent with the epidemiological trends of obesity that have steadily increased in all global regions [49].

The rationale for this study to measure clinical outcomes of adiposity by FMI and not only OW/OB through BMI is given by the importance of understanding the relevant differences of these two surrogates of adipose tissue accumulation. It is important to note that we do not have reference values of BMI, and almost 5% of subjects classified as healthy weight by BMI had FMI values above the mean of their OW/OB peers, which is a potential misclassification.

The human body is composed by four major compartments: bone mineral content, total body water, fat mass and lean mass; of these compartments, fat mass or adipose tissue has been consistently related to metabolic and inflammatory activities responsible for negative health outcomes (e.g., diabetes mellitus, hypertension, metabolic and cardiovascular risks) [10,50]. Currently, subjects may be classified as OW/OB according to specific BMI cut-off values that are supposed to estimate the pathological accumulation of adipose tissue [51]. Nevertheless, BMI is not able to inform the distribution of weight across such compartments and is prone to misclassification. Subjects with low lean mass and high adipose tissue may be classified as “healthy” weight because of their normal BMI values, representing a potential risk not being detected (in our data, 6% of subjects classified as healthy weight by BMI had FMI values above the mean of OW/OB peers). Conversely, we also observed that 3% of subjects classified as OW/OB by BMI had FMI values below the mean of “healthy” peers (data not shown), representing subjects with high lean mass and low adipose tissue being wrongly classified as OW or OB. Although BMI is a simple, practical, and easy to use clinical tool, acknowledgment of its potential misclassification is important, and, where feasible, other measurements may better inform about the current and future risks associated with the pathological accumulation of fat mass. Similar to our results, Costa-Urrutia et al. have recently reported that between 8% and 17% of Mexican children and adolescents may show healthy BMI values, although they may have an excess of bioimpedance-based adiposity, meaning males with >25% and females >30% of percent body fat. In contrast, we described adiposity by means of FMI measured by DXA, and considered possible misclassification according to the proportion of subjects categorized as healthy according to BMI values but showing FMI values equal to or above their overweight or obese peers’ means [13].

## 5. Limitations

Some limitations of this study should be noted. First, the use of lifestyle habits questionnaires as a main source of data collection can be considered a limitation: the weaknesses of using these types of questionnaires due to misreporting have been described in similar studies. Misreporting is associated with different types of bias, such as: social desirability bias, recall bias and selection bias [52,53]. This could certainly hamper accurate estimations of the influence of such habits on adiposity. Even when the questionnaires have been validated in our population, there are other methods, such as double-labeled water, podometers and accelerometers, that could be used to minimize this problem. Unfortunately, these instruments are not available in our setting, however, for large epidemiological studies, in the context of developing countries, short, simple, practical and validated questionnaires are considered adequate [27,54]. Still, to evaluate such bias we applied the Goldberg cut-off method [26] and presented the percentages of under-reporting (3%), over-reporting (15%) and plausible reports (81%). These proportions of misreporting are similar to what has been previously described in other epidemiological studies. Mensink et al., in a multinational European study, reported rates between 0.5% and 5% of underreporting in children aged 4 to 10 years old, and from 0.6% to 34% in adolescents of 11 to 17 years old [55]. Another review of 28 cross-sectional studies, reported proportions from 2% to 85% of underreporting, and from 3% to 46% of over-reporting [56]. Nevertheless, applying the Goldberg cut-off method informed us regarding the mean bias of our sample. To describe the effect of these biases, we analyzed all associations between OL components with FMI and OW/OB, using the total sample as well as a plausible-report (81%) subsample with no significant differences between them. Following the principle of parsimony, we reported the results of the total sample. The routine clinical and nutritional assessment of the pediatric population in Mexico relies on similar methods; our data collection may be closer to the reality of such clinical settings, and therefore, we believe that the interpretability and applicability of our findings have a robust external validity and may be valuable.

Another potential limitation is whether our results could be generalized to the whole population of children and adolescents in Mexico. We included a large sample of subjects from the Metropolitan Area of Mexico City. Mexican subjects from rural areas and indigenous populations, as well as other territories of Mexico, may not share the same characteristics as our sample. The Metropolitan area of Mexico concentrates almost a fifth of the total population of the country. Many of the people living in this area have migrated from other States or other areas of Mexico; therefore, we believe that a reasonable representation is achieved with this sample. However, we believe that future studies should address whether the prevalence of OL components and their associations with adiposity, overweight and obesity are like ours. Other studies have identified relevant differences among different ethnic groups living in the same territory, and thus genetic and cultural background may be seen as relevant cofounding factors, not addressed in the present study [34].

Measuring fat mass with DXA instead of using only BMI gave strength and innovation to our study, but it may be difficult to replicate given its low availability and the technical difficulties involving the pediatric population, making it not feasible, expensive and unpractical. To address this, we also reported our results in terms of BMI and risk associations for OW/OB, which may be easier to transfer to routine clinical practice. 

Other limitations were that some known confounders for OW/OB and adiposity (e.g., genetic background, socioeconomic background, mother anthropometric variables, conditions of pregnancy and birth, and psychological factors) were not explored, and thus their influence in our associations is not known [57,58].

## 6. Conclusions

In summary, a very high prevalence of OL components with a significant risk association to increased adiposity, overweight and obesity affects Mexican children and adolescents from the Metropolitan Area of Mexico City. Given the fact that OW/OB are major health issues in Mexico, there is an urgent need for high-level actions. Health policies should focus on the promotion and facilitation of the healthy habits that should be acquired and integrated in the environment and behavior of the population. Focus on three specific actions—improving quality of diet, increasing physical activity and decreasing sedentarism—may help revert the trends for these health threats.

## Figures and Tables

**Table 1 nutrients-12-00819-t001:** Characteristics and obesogenic lifestyle (OL) components in Mexican children and adolescents.

Variable	Children			Adolescents		
Sex	Female	Male	*P*	Female	Male	*P*
	349 (44%)	451 (56%)		309 (48%)	340 (52%)	
Age (years)	9.0 ± 1.7	9.0 ± 1.6	0.73	14.9 ± 1.7	14.7 ± 1.7	0.17
Weight (kg)	32.4 ± 11.4	32.8 ± 11.0	0.60	56.4 ± 12.0	57.6 ± 14.4	0.24
Height (cm)	131.9 ± 12.1	132.3 ± 11.1	0.66	156.2 ± 6.4	163.7 ± 9.8	<0.001
Tanner puberal stage			<0.001			<0.001
*I*	211 (60%)	382 (85%)		0 (0%)	16 (5%)	
*II*	95 (27%)	64 (14%)		7 (2%)	45 (13%)	
*III*	37 (11%)	4 (1%)		65 (21%)	86 (25%)	
*IV*	6 (2%)	1 (0%)		145 (47%)	118 (35%)	
*V*	0 (0%)	0 (0%)		92 (30%)	75 (22%)	
**Habit**						
Physical Activity (min/day)	17.9 ± 21.4	25.1 ± 24.8	<0.001	16.8 ± 23.7	31.8 ± 29.4	<0.001
Screen Time (h/day)	3 ± 1	3 ± 2	0.04	4 ± 2	4 ± 2	0.13
Sleep time (h/day)	9 ± 1	9 ± 1	0.07	8 ± 2	8 ± 1	0.11
Daily diet intake						
Energy/day (kcal)	2215 ± 611	2436 ± 724	<0.001	2375 ± 822	2926 ± 857	<0.001
Energy/day (kJ)	9272 ± 2559	10,397 ± 5220	<0.001	10,084 ± 4229	12,249 ± 3587	<0.001
Carbohydrates (g)	301 ± 93	337 ± 116	<0.001	335 ± 222	400 ± 135	<0.001
Proteins (g)	86 ± 29	98 ± 111	0.07	92 ± 39	127 ± 233	0.01
Fat (g)	76 ± 30	86 ± 60	0.01	85 ± 57	104 ± 79	<0.001
Fiber (g)	23 ± 11	23 ± 11	0.31	23 ± 12	26 ± 12	<0.001
AHA Diet score	1.6 ± 1	1.5 ± 1.1	0.21	1.4 ± 1	1.3 ± 1	0.13
Estimated Basal Metabolic Rate*	1107 ± 165	1222 ± 212	< 0.001	1401 ± 120	1674 ± 241	<0.001
Estimated Energy Expenditure**	1709 ± 365	1938 ± 465	< 0.001	2254 ± 303	3018 ± 607	<0.001
EI reported: EE estimated***	1.34 ± 0.42	1.31 ± 0.44	0.029	1.08 ± 0.41	1 ± 0.33	0.012

Abbreviations: AHA: American Heart Association. Data are shown as mean ± SD or *n* (%). Differences that reached statistical significance (*P* value < 0.05) by unpaired *t*-tests for continuous variables and chi-squared tests for categorical variables. *BMR: basal metabolic rate calculated from Schofield equations.**Energy expenditure = BMR x physical activity level, ***EI reported: EE estimated: ratio between energy intake reported by estimated energy expenditure.

**Table 2 nutrients-12-00819-t002:** Adiposity measurements in Mexican children and adolescents.

Adiposity	Children			Adolescents		
	Female *n* = 349 (44%)	Male *n* = 451 (56%)	*P*	Female *n* = 309 (48%)	Male *n* = 34 (52%)	*P*
Total Fat Mass (kg)	11.3 ± 6.0	10.6 ± 6.2	0.136	21.1 ± 7.5	15.0 ± 8.1	<0.001
Trunk Fat (kg)	5.1 ± 3.4	4.7 ± 3.5	0.93	10.5 ± 4.4	7.2 ± 4.8	<0.001
Android Fat (kg)	0.7 ± 0.6	0.7 ± 0.6	0.298	1.6 ± 0.8	1.1 ± 0.9	<0.001
Gynoid Fat (kg)	1.8 ± 0.9	1.6 ± 0.9	0.008	3.7 ± 1.3	2.4 ± 1.3	<0.001
Appendicular Fat (kg)	5.5 ± 2.6	5.2 ± 2.7	0.145	9.9 ± 3.2	6.9 ± 3.3	<0.001
Fat percentage (%)	33.5 ± 7	30.5 ± 7.9	<0.001	36.8 ± 6.2	25.1 ± 8.6	<0.001
FMI (kg/m2)	6.3 ± 2.6	5.8 ± 2.8	0.027	8.6 ± 2.9	5.6 ± 2.9	<0.001
BMI (kg/m^2^)	18.1 ± 3.7	18.3 ± 3.9	0.52	23.0 ± 4.2	21.3 ± 4.2	<0.001
BMI categories			0.50			<0.05
Underweight	17 (5%)	22 (5%)		2 (1%)	23 (7%)	
Normal weight	222 (64%)	278 (62%)		193 (62%)	223 (66%)	
Overweight	55 (16%)	62 (14%)		73 (24%)	44 (13%)	
Obesity	55 (16%)	89 (20%)		41 (13%)	50 (15%)	

Abbreviations: BMI body mass index, FMI fat mass index. Data are shown as mean ± SD, or number (%). Comparisons between sex groups where done by unpaired-t test (for continuous variables) and chi-squared tests (for categorical variables).

**Table 3 nutrients-12-00819-t003:** Means and standard deviation of fat mass index (FMI) (kg/m2) according to presence or absence of each OL component in Mexican children and adolescents.

Age Group	Sex	Physical Activity					Screen Time					Sleep Duration					AHA Diet score**						
		Active		Inactive			Adequate		Excessive			Adequate		Insufficient			Healthy		Intermediate		Poor		
		*n* (%)	*Mean ± SD*	*n* (%)	*Mean ± SD*	*P*	*n* (%)	*Mean ± SD*	*n* (%)	*Mean ± SD*	*P*	*n* (%)	*Mean ± SD*	*n* (%)	*Mean ± SD*	*P*	*n* (%)	*Mean ± SD*	*n* (%)	*Mean ± SD*	*n* (%)	*Mean ± SD*	*P***
**Children**	F	27 (8%)	4.6 ± 1.9	322 (92%)	6.4 ± 2.6	0.001*	188 (54%)	6.0 ± 2.5	161 (46%)	6.6 ± 2.6	0.02*	241 (69%)	6.2 ± 2.4	108 (31%)	6.4 ± 2.9	0.50	12 (3%)	5.3 ± 1.3	161 (47%)	6.4 ± 2.7	170 (50%)	6.2 ± 2.5	0.20
	M	64 (14%)	5.3 ± 2.5	387 (86%)	5.9 ± 2.8	0.09	228 (51%)	5.5 ± 2.4	223 (49%)	6.2 ± 3.1	0.01*	338 (75%)	5.7 ± 2.6	113 (25%)	6.3 ± 3.2	0.03*	17 (4%)	5.3 ± 2.6	198 (45%)	5.5 ± 2.6	229 (52%)	6.1 ± 2.9	0.53
**Adolescents**	F	29 (9%)	8.3 ± 2.9	277 (91%)	8.6 ± 2.8	0.50	83 (27%)	8.4 ± 2.9	223 (73%)	8.7 ± 2.8	0.51	177 (58%)	8.5 ± 3.0	129 (42%)	8.8 ± 2.6	0.29	4 (1%)	8.5 ± 4.3	133 (44%)	8.8 ± 2.8	165 (55%)	8.4 ± 2.8	0.93
	M	84 (25%)	4.8 ± 2.5	256 (75%)	5.9 ± 3.0	0.004*	105 (31%)	5.7 ± 2.9	235 (69%)	5.6 ± 2.9	0.80	218 (64%)	5.5 ± 2.9	122 (36%)	5.7 ± 2.9	0.47	7 (2%)	5.9 ± 2.6	132 (39%)	5.9 ± 3.1	196 (59%)	5.3 ± 2.7	0.76

Abbreviations: F = female, M = male. Healthy diet score (5–4 points), Intermediate (3–2 points), Poor (1–0 points). * Differences that reached statistical significance (*P* value < 0.05) by unpaired-t tests. **Mean differences assessed by ANOVA, comparison with healthy diet.

**Table 4 nutrients-12-00819-t004:** Risk association for overweight/obesity (OW/OB) according to the presence or absence of each OL component in Mexican children and adolescents.

Age Group	Sex	Inactive			Excessive Screen Time			Insufficient Sleep			Unhealthy Diet		
		OR	95% CI	*P*	OR	95% CI	*P*	OR	95% CI	*P*	OR	95% CI	*P*
**Children**	All	2.03	(1.19, 3.50)	0.008	1.12	(0.83, 1.50)	0.45	1.27	(0.91, 1.77)	0.16	1.10	(0.44, 2.70)	0.83
	F	6.55	(1.52, 28.22)	0.004	0.91	(0.57, 1.44)	0.69	1.04	(0.628, 1.70)	0.89	2.28	(0.48, 10.74)	0.28
	M	1.50	(0.83, 2.72)	0.176	1.30	(0.87, 1.93)	0.19	1.50	(0.96, 2.36)	0.07	0.62	(0.187. 2.07)	0.44
**Adolescents**	All	1.57	(0.99, 2.50)	0.05	1.06	(0.73, 1.54)	0.74	1.21	(0.87, 1.70)	0.26	2.34	(0.66, 8.30)	0.17
	F	0.70	(0.32, 1.50)	0.36	0.89	(0.53, 1.50)	0.66	1.40	(0.84, 2.16)	0.21	2.10	(0.43, 10.30)	0.35
	M	2.20	(1.20, 4.10)	0.012	1.23	(0.72, 2.10)	0.44	1.02	(0.62, 1.70)	0.91	2.90	(0.40, 24.30)	0.30

Simple univariate risk association stratified by sex and age groups. OR = odds ratio, CI = confidence interval.

**Table 5 nutrients-12-00819-t005:** Relationships between OL cumulative score with FMI (kg/m2) in Mexican children and adolescents.

Age Group	Sex		OL Cumulative Score											Effect of OL Cumulative Score on FMI		
		0 OL Components		1 OL Component		2 OL Components		3 OL Components		4 OL Components						
		*n* (%)	FMI (kg/m2) mean ± SD	*n* (%)	FMI (kg/m2) mean ± SD	*n* (%)	FMI (kg/m2) mean ± SD	*n* (%)	FMI (kg/m2) mean ± SD	*n* (%)	FMI (kg/m2) mean ± SD	**P*		beta	95% CI	***P*
Children			3.6 ± 0.5		5.2 ± 2.1		5.8 ± 2.4		6.2 ± 2.9		7.0 ± 3.2	<0.001		0.54	(0.32, 0.75)	< 0.001
	F	0	-	35 (10%)	5.2 ± 1.8	134 (38%)	6.0 ± 2.4	136 (39%)	6.5 ± 2.7	44 (13%)	7.0 ± 2.9	0.008		0.55	(0.24, 0.87)	0.001
	M	4 (1%)	3.6 ± 0.5	51 (11%)	5.2 ± 2.2	185 (41%)	5.7 ± 2.4	163 (36%)	5.9 ± 3.0	48 (11%)	6.9 ± 3.4	0.01		0.50	(0.21, 0.80)	0.001
Adolescents			6.2 ± 2.0		6.3 ± 3.2		6.7 ± 3.2		7.2 ± 3.4		7.5 ± 2.9	0.08		0.41	(0.13, 0.69)	0.004
	F	1 (0.3%)	6.7 ± NA***	14 (5%)	9.0 ± 3.1	76 (25%)	8.2 ± 2.9	141 (46%)	8.7 ± 3.1	74 (24%)	8.8 ± 2.3	0.625		0.19	(-0.19, 0.57)	0.333
	M	2 (1%)	5.9 ± 2.7	36 (11%)	5.2 ± 2.6	105 (31%)	5.6 ± 3.0	136 (40%)	5.6 ± 3.0	61 (18%)	5.9 ± 2.8	0.89		0.16	(-0.18, 0.50)	0.356

* ANOVA, adjusted for multiplicity Bonferroni (*p* < 0.05). Post-hoc analyses showed that the FMIs of female children with three and four OL components (bold values) were significantly higher compared to the FMIs of female children with one OL component. Male children with four OL components (bold values) showed significantly higher FMI values when compared to those with 0 or one OL component, and male children with three OL components (bold values) showed significantly higher FMI values compared to those with 0 OL components. ** Simple regression analysis stratified by age and sex groups. *** Category with less than two cases.

**Table 6 nutrients-12-00819-t006:** Multiple regression analysis of OL components to FMI (kg/m2) stratified for sex and age groups and adjusted for Tanner.

Age Group	Sex	Inactivity			Excessive Screen Time			Insufficient Sleep			Unhealthy Diet			Tanner Stage		
		Beta	95% CI	**P*	Beta	95% CI	**P*	Beta	95% CI	**P*	Beta	95% CI	**P*	Beta	95% CI	**P*
Children	F	1.78	(0.81, 2.80)	<0.001	0.5	(−0.02, 1.0)	0.057	−0.02	(−0.6, 0.55)	0.90	0.97	(−0.51, 2.40)	0.20	0.85	(0.50, 1.20)	<0.001
	M	0.68	(−0.60, 1.4)	0.07	0.65	(0.13, 1.17)	0.014	0.62	(0.02, 1.20)	0.04	0.49	(−1.16, 2.15)	0.50	0.02	(−0.61, 0.65)	0.95
Adolescents	F	0.30	(−0.79, 1.38)	0.60	0.04	(−0.68, 0.77)	0.90	0.23	(−0.42, 0.88)	0.50	−0.013	(−1.90, 1.87)	0.90	0.65	(0.23, 1.07)	0.002
	M	1.02	(0.32, 1.72)	0.005	0.21	(−0.48, 0.90)	0.50	0.34	(−0.30, 0.98)	0.30	0.08	(−1.92, 2.09)	0.90	−0.48	(−0.77, −0.20)	0.001

*P* values of multiple regression analyses stratified by age and sex groups and adjusted for Tanner. Significant associations are shown in bold. * Differences that reached statistical significance (*P* < 0.05).

**Table 7 nutrients-12-00819-t007:** Risk association for OW/OB according the cumulative effect of OL components in Mexican children and adolescents.

Age Group	Sex	0 or 1 OL Component (Reference Group)	2 OL Components				3 OL Components				4 OL Components			
		OW/OB *n* (%)	OW/OB *n* (%)	OR	95% CI	*P*	OW/OB *n* (%)	OR	95% CI	*P*	OW/OB *n* (%)	OR	95% CI	*P*
**Children**	F	4 (12%)	47 (36%)	4.3	(1.4, 12.8)	0.01	45 (36%)	4.17	(1.4, 12.6)	0.011	14 (33%)	3.75	(1.1, 12.8)	0.03
	M	16 (30%)	59 (32%)	1.1	(0.56, 2.1)	0.77	54 (35%)	1.22	(0.6, 2.4)	0.55	22 (48%)	2.1	(0.9, 4.8)	0.07
Adolescents	F	6 (40%)	26 (35%)	0.8	(0.25, 2.48)	0.69	55 (39%)	0.96	(0.3, 2.8)	0.94	26 (35%)	0.81	(0.3, 2.5)	0.72
	M	9 (24%)	28 (28%)	1.2	(0.5, 2.89)	0.66	37 (29%)	1.25	(0.5, 2.9)	0.6	20 (36%)	1.7	(0.7, 4.4)	0.24

Risk associations based on a comparison of groups with two, three or four OL vs. the reference group of subjects with 0 or one OL components.

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
