# Peer review of "Obesogenic Lifestyle and Its Influence on Adiposity in Children and Adolescents, Evidence from Mexico"

_nutrients, 2020, doi:10.3390/nu12030819_

Round 1

Reviewer 1 Report

Please see an attachment.

Reviewer 2 Report

The manuscript submitted by Lopez-Gonzalez et al describes the obesogenic lifestyle risk factors (OL) responsible for overweight and obesity in children and adolescents living in Mexico City. They show that the obesity epidemic is due to three risk factors explaining the high BMIs and fat mass indexes: malnutrition with too low fibre and fish intakes, inactivity through leisure taken in front of a screen and lack of moderate to strong physical activity. They conclude by suggesting that the Mexican authorities must play on these 3 risk factors to fight against the high obesity prevalence encountered in the Mexican population in general.

The study seems to have been done carefully. One can regret that some data were obtained through a questionnaire and were not measured directly (sleep duration with adequate materials, physical activity and sedentariness with i-phone and adapted software). However, we recognize that these materials are not easily available for a so high sample size. Although the results are not surprising, I think that they merit to be published in order to alert the Mexican authorities and to drive them in the fight against this nuisance. Another point is regrettable in this study: the calculation made did not allow the classification of the risk factors. This would been interesting to establish priorities in the fight against the disease. However, this point was correctly discussed in the paragraph describing the limitations of the study.

Author Response

Response to reviewer 2:

Thank you very much for your comments.

Round 2

Reviewer 1 Report

Despite my repeated recommendations, many concerns remain in the present manuscript and some comments were neglected by the authors. Further, the authors report where they have made revisions but almost all line numbers are shifted. Please revise the manuscript more carefully.

Author Response

We have carefully reviewed and changed the manuscript accordingly your last comments received March 3, 2020. All changes appear in this new revisited version in red, not to be confused with older versions.

Unfortunately we think that there was some misunderstanding in the previous revisions probably due to some errors within the lines. In this new version we have been over all lines and topics and hopefully all your comments and suggestions are covered satisfactory.

In order to keep the lines in order probably is better to read the PDF version instead of the word version.

We thank you in advance for all the time and input given to our work

  1. The results in line 253-274 are not shown as table(s) and Table 6 is not available in the present manuscript. Again, I strongly recommend doing that. If not, the authors should explain the reason why they do not. Further the results in line 236-239 (Table 4) should be shown in each sex and age group in line with other results.

Results from lines 253-274 has been added to tables 5, 6 and 7.

Table 6 added

Table 5 is referenced in line 334

Table 6 is referenced in line 339

Table 7 is referenced in line 347

  1. Please clearly state where the authors have revised. At least, the results in line 253-265 are not analyzed by age group and the results in line 267-270 are not necessary. The results in line 271-274 should be shown by sex and age groups.

Results from line 253-265 now appear in table 5 and the ones from lines 267-270 are in table 7. Tables are referenced as mentioned in point 1

  1. The authors have not answered my comment. Is there any previous study investigating the association between OL components and fat mass? If there are such studies, please compare the present results with their results. If not, the authors should state so. Without the statement, line 311-313 is non-sense because the authors can choose studies that do not include Mexican population and fat mass measurement.

Yes, there have been previous studies with the association of OL components and fat mass and were compared accordingly with your suggestion. Comments appear in the following lines: 422-435, 444-452, 473-483 and 486-498.

  1. Line 420-424 may mention the mechanism. However, this section is too short to explain the underlying mechanism although the association between OL components and fat mass is main outcome in the present study. Please add more explanation of the influence of each OL as well as its interaction.

This observation are taken into account (discussion sections added in point 3)

  1. On the other hand, the section comparing the prevalence of OB/OW as well as OL components between the present and previous studies should be condensed. This section is still long. Please reduce this section less than 1 page.

Changes are done

  1. If the authors remain the results as Figure 1, they should show the results by age and sex groups in line with other results. Additionally, they should add rationale for investigating the association in the introduction section and the discussion about this issue. Although line 56-58 may be the rationale, the authors should state that they have added these sentences as the rationale and the reference should be needed. Further, the previous Table 4 remains as a supplementary table in contrast to the authors’ response. Please eliminate supplementary table because there is no description about it in the present manuscript.

Since confusing and not the primary objective of this communication we withdraw figure 1 and table 4 at the supplementary material

Introduction

  1. Please describe the reason for using fat mass as a marker of obesity additionally to BMI.

I could not find these sentences in the introduction section. Anyway, the authors state where the authors have added these sentences to the present manuscript.

Find the changes in lines 59 and 60

Methods

  1. In contrast to my comment, line 65-66 is not included in the section 2.1. Again, I recommend the authors doing so. Further, the information on the procedure of recruitment is insufficient. Please show the number of schools where the participants were recruited as well as the number of students asked to participate in the present study; the latter (but not 1,767 participants interested in line 185) are the eligible participants. Therefore, the present response rate (82% in line 188) is irrelevant. To move the line 185-188 to line 79 may helpful in understanding the procedure of recruitment. At least, the authors should clearly state that they have added line 185-188.

Your comments were taken into account. Number of schools and recruitment methods are added lines 73-78, 83 to 85

  1. The authors do not show where they have made the revision. Further, they have made wrong change. Please locate line 148-151 (line 82-86 in the previous manuscript) after line 89 in the present manuscript (I don’t know why the authors have moved this section without my suggestion) and then, locate line 140-146 (line 105-110 in the previous manuscript) after that. In accordance with this change, the title 2.2.1 should be changed to 2.3 and the title 2.2.2. should be deleted.

We think that some problems in the past manuscripts are due to numbering. Since only the line numbers not the topic is mentioned, quite confusing over the past manuscripts.

We have read the manuscript carefully for this revision and find that, the information for every section is correct.

Adiposity measures are in line 101 to 106 and BMI classification in 108 to 111 as you wanted in one of the previous manuscripts

  1. Line 87-93

Please detail dietary intake estimation using both 24-h recall and 12-month FFQ. Further, the authors should mention how they controlled energy misreporting because energy misreporting is common problem in self-reported dietary intake and can hamper the accurate estimation of influence of OL components on OB/OW and fat mass.

The authors should state where they have made the revision in the sections of methods and limitations. Additionally, only line 123-126 is sufficient to detail dietary intake estimation. Other sentences added to the section 2.2.3 in the present revision should be eliminated because I did not suggest adding these sentences and no explanation is available in the authors’ response. The procedure that the authors mentioned above (in our study…. and sweetened beverages) is insufficient to control energy misreporting. Therefore, please add the ratio of energy intake to basal metabolic rate or the ratio of energy intake to estimated energy requirement to Table 1 (or in the text), as a measure of energy misreporting. Depending on the degree of energy misreporting, the influence of the present results should be appropriately discussed in the discussion section.

Attending your comments the misreporting bias is described in lines 164-169, and supplementary tables 1 and 2 containing these information are included.

Additionally regarding the same topic, results appear in lines 247 -251

  1. The authors measured screen time not sitting or lying time. Additionally, screen time is considered to be more obesogenic because of the influence of advertisements of energy-dense food items. Therefore, “sedentary” should be revised as “screen time”.

Thank you for your revisions.

  1. The section of statistical analysis is line 152-182 and the description is still insufficient. First the authors should compare general characteristics between sexes in each age group by t-test and chi-square test (Table 1) in line with the results in Table 2 and describe the procedure. Line 160-162 is irrelevant because these values were compared between sexes. Line 164-165 is also insufficient because the comparison in Table 3 is conducted in each sex and age group the comparison across AHA score is conducted using ANOVA. For line 165 to 167, ORs (95%CIs) should be estimated in each sex and age groups in accordance with my comment 1 and the method (logistic regression?) should be stated. Line 167-168 should be eliminated in accordance with my comment 6. I will comment to line 170-179 after the authors provide tables of the results from analysis in this section.

Changes done, lines 185-186, 194-195, 197, 200 -201, 203 and 215

We kept Bonferroni test, justification appears in line 201

  1. Moreover, please state the software used for statistical analysis and the alpha level for significance.   This information may be shown in line 181-182. For the alpha level for significance, the authors just have to mention it only once in the first or last part of this section.

This information appears in line 214-215

Results

  1. The revisions are insufficient. Again, I recommend the authors showing the number and % (as well as means) of participants “in each sex and age group“ in the results section (i.e., line 195-223 and line 284-285). However, the results in line 188-189 should be shown as number (e.g. In total, XXX children aged XX to XX years (XX boys and XX girls) and XXX adolescents aged XX to XX years (XX boys and XX girls) are included the present analysis.)

Done, lines 218-219

  1. The changes are insufficient. Again, please include line 185-191 in the section 3.1 in accordance with my previous comment (although line 185-188 may move to the methods section in accordance with my comment 8). The means of BMI still remain in Table 1 and thus, the authors should move them to Table 2. Further, the prevalence of overweight and obesity (line 131-133 in the previous manuscript) is deleted although it is one of main outcomes in the present study.

This data appear in lines 276-282 as well as in table 2

  1. In contrast to the authors’ comment, line 195-199 remains. Please delete this section. Although I suggested to add “the mean” of AHA score to Table 1 in line with other components, the authors have added “its prevalence”. Again, I recommend the authors adding its mean of AHA score to Table 1. Supplemental Table shows the association between BMI classification and adiposity but not the results of the previous Table 2.

Line 195-199 is the footnote of table 1. Now placed in the table.

AHA score is added as well in table 1

17. Please clarify the statistical method used to compare fat mass between sexes in the method section. This information may be shown in line 158-162, but the description is insufficient (see my comment 12).

Line 185-186 Statistical method used was unpaired t-test

18. This section may be shown in line 230-233 and line 242-250.

Section moved to lines 282-285 (adiposity description)

19. Line 226 in the present manuscript

Please state which participants are compared with female adolescent

This comment was not clear since no comparison is referred to. Only a description of the OL components

Discussion

20. Line 227-229. These sentences should be moved to the introduction section.

Thank you for your revisions.

21. This section may be shown in line 465-471.

We feel that this paragraph should be kept in conclusions and appear in lines 573 to 576

22. The information is shown in line 427-461. Although the authors have added more limitations, they should add appropriate references and discuss the influence of each limitation on the present results. Further, the description in line 439-442 is not necessary. Moreover, the influence of residual confounding should also be mentioned.

This section has been expanded adding the references accordingly

23. Tables

The changes are insufficient and please make sure the decimal places in the tables are uniform within the same variables as well as p-values. I suggest the authors revising tables as follows:

Table 1:

The title should be changed (e.g., General characteristics and OL components in Mexican children and adolescents). “N = 1449” as well as n and % of total children and adolescents (800, 52% and 649, 48%) should be removed because the readers can calculate these values. The prevalence of Tanner puberal stages for total participants (609, 42.0% to 167, 11.5%) should also be removed. For AHA score, its mean values (not but its prevalence) should be shown. The order of other OL components should be sorted in line with the text (i.e., PA, screen time, and sleep). Please add p-values for the comparison of each variable between sexes to the table and show the statistical method used in footnote.

Changes done accordingly with your suggestions

Table 2:

The title should be changed to “Adiposity measurements in Mexican children and adolescents.” N and % of total children and adolescents (800, 52% and 649, 48%) as well as BMI classification for total participants (916, 63.2% to 64, 4.4%) should be removed. For BMI classification, only one P-value should be shown in each age group (two P-values in total) because the authors may conduct chi-square test. Footnote should be revised (e.g., Values are expressed as “means±SD or n (%)”. Comparisons where done by unpaired-t “(for continuous variables)” and chi-squared tests “(for categorical variables)”). Please move the mean BMI from Table 1 to Table 2.

Changes done accordingly with your suggestions

Table 3:

The title should be changed to “Means and standard deviation of FMI (kg/m2) according to presence or absence of each OL component in Mexican children and adolescents.” Abbreviations for females to poor should not be used because there is enough space in the table and readers can’t distinguish “P-values” and “poor.” Please add n of each OL component group in each sex and age group. Because the readers can understand differences that reached statistical significance by P-values, * should be shown with P of PA, screen time, and sleep (i.e., P*). In accordance with this change, its footnote (Difference……) should be revised. Similarly, ** should be shown with P of AHA score (i.e., P**). Its footnote should be revised because adjustment for multiplicity is not necessary for ANOVA. If the authors conduct multiple comparison across AHA score categories, they should state which categories are compared with (e.g., health vs. intermediate).

Changes done accordingly with your suggestions

Table 4:

The title should be changed to “Risk association for OW/OB according to the presence or absence of each OL component in Mexican children and adolescents.” If the authors conduct multiple analysis (adjustment for Tanner puberal stages?), the result should be shown instead of univariate analysis. Further, the results should be shown in each sex and age group in line with other tables. Please add n of each sex and age group as well as prevalence of OW/OB participants in presence and absence group for each OL component. The statistical method used should be shown in the footnotes. For diet, the authors should state categories are compared with because there are three categories.

Only simple associations are presented in this table adjusted by sex and age group.

Adjustment for Tanner pubertal stages is in table 6.

Table 5:

The title should be changed to “Relationships between OL cumulative score with FMI (kg/m2) in Mexican children and adolescents.” If the authors conduct multiple analysis (adjustment for Tanner puberal stages?), the result should be shown instead of univariate analysis. The headings of “FMI (kg/m2) mean±SD” as well as “*P, beta, 95%CI, and **P-Value” should be include in each heading of “OL cumulative score” and “Effect of OL cumulative score on FMI”, respectively. Please add n of each OL cumulative score group in each sex and age group. For female adolescents, the first and second groups for OL cumulative score should be combined for performing ANOVA because of the sparseness of participants. For P-values for ANOVA, actual values (but not <0.05 or NS) should be shown. Its footnote should be revised because adjustment for multiplicity is not necessary for ANOVA. If the authors conduct multiple comparison across AHA score categories, they should state which categories are compared with (e.g., the first vs. the second groups). The explanation about stratification in footnote for generalized linear model is unnecessary.

No adjustment was done by tanner puberal stages in this table

We use Bonferroni since we wanted to identify the different group.

All p values were added and headings corrected.

Figure 1: The results should be shown in each sex and age group in line with tables.

Figure 1 is eliminated.

This manuscript is a resubmission of an earlier submission. The following is a list of the peer review reports and author responses from that submission.

Round 1

Reviewer 1 Report

Although the topic is potentially important and interesting, there are several concerns, which should be sufficiently addressed to improve the quality of the manuscript.

First of all, the authors should show the results in line 202-225 as table(s) (that should also include the results in Table 5) because the results are main outcome of the present study. Further, please revise the title (e.g. Associations between OL components and adiposity) because it is difficult to understand the contents of this section by the present title. Further the analysis in line 217-225 should be conducted by age groups in accordance with other analysis.

I think the novelty of the present study is estimating the association between OL components and adiposity using fat mass. Nonetheless, comparison of the association in the present study with that in the previous studies as well as discussion of the reason for the difference or consistence are unavailable in the discussion section. This is another measure limitation of the present study. Therefore, the authors should discuss these issues. Additionally, the authors should explain the underlying mechanism that derives the association between OL components and fat mass. On the other hand, the section comparing the prevalence of OB/OW as well as OL components between the present and previous studies (line 239-337) should be condensed.

The association between fat mass and BMI (line 166-175 and 191-196) may not necessary. This is because there is no rationale for investigating the association in the introduction section and the discussion about this issue is insufficient. At least, these sections should be combined and moved to the last part of the results section as section 3.4. Similarly, the results in Table 4 and Fig 1 should be shown as one table or figure, and the table (or figure) should be shown as the last table or a supplementary material.

Introduction

Please describe the reason for using fat mass as a marker of obesity additionally to BMI.

Methods

Line 67-76

Please add the title to this section (e.g., 2.1. Study participants). Moreover, the authors should describe the procedure of recruitment, such as, the number of schools where participants were recruited, sampling method (i.e., random sampling or voluntary), the eligible number of participants, and the number of participants excluded from study because of missing information.

Line 105-110

This section should be located after line 86.

Line 87-93

Please detail dietary intake estimation using both 24-h recall and 12-month FFQ. Further, the authors should mention how they controlled energy misreporting because energy misreporting is common problem in self-reported dietary intake and can hamper the accurate estimation of influence of OL components on OB/OW and fat mass.

Line 113

The authors measured screen time not sitting or lying time. Additionally, screen time is considered to be more obesogenic because of the influence of advertisements of energy-dense food items. Therefore, “sedentary” should be revised as “screen time”.

Line 118-127

The description about statistical analysis is insufficient to understand that which groups were compared and what covariates were adjusted in each analysis. Please revise this section. Moreover, please state the software used for statistical analysis and the alpha level for significance.

Results

The number and % of participants should be shown in each sex and age group because the analysis was conducted by sex and age groups. Further, “CI95%” must be “95%CI.”

Line 129-133

Please include this section in the section 3.1. The response rate in line 133 should be located after the first sentence. The prevalence of overweight and obesity (line 131-133) should be shown in the section 3.2 because these variables were the outcome rather than characteristics. Thus, the values of BMI and its classification in Table 1 should be moved to Table 3.

Line 135-160

Line 135-138 should be removed because the same information is shown in following parts (line 143-190). Moreover, line 138-140 should be moved to the last part of this section. This information should be shown after the description about each component of OL. Please revise “food intake analysis” in line 140 as “energy and macronutrient intakes” and “Calories” in Table 1 as “Energy“. Further, please add the mean of AHA Healthy Diet Score to Table 1 in accordance with other components and show Table 2 as a supplementary material.

Line 162-164

Please clarify the statistical method used to compare fat mass between sexes in the method section.

Line 176-179 and 183-190

Please include these sections in the section 3.3.

Discussion

Line 227-229

These sentences should be moved to the introduction section.

Line 344-348

These sentences should be moved to conclusion.

Line 349-351

There are more limitations should be mentioned, such as, generalizability of the present results, the validity of questions used to collect information on OL components, and the influence of energy misreporting on the present results.

Tables

Please remove vertical line in Tables 2, 4, and 5, and add the number of participants to Tables 2-5. Further, please explain the statistical method used in the footnotes and add P-values to Tables.

Author Response

Thank you for your valuable observations and comments; they would be useful to improve the manuscript.

Below you could find your comments followed by our answers in blue. Hopefully you will be satisfied with the changes.

First of all, the authors should show the results in line 202-225 as table(s) (that should also include the results in Table 5) because the results are main outcome of the present study. Further, please revise the title (e.g. Associations between OL components and adiposity) because it is difficult to understand the contents of this section by the present title.

The table 5 now are present as table 3, and we add table 4 and 6 with the results in line 202-225. Also the title of the table was changed.

2. Further the analysis in line 217-225 should be conducted by age groups in accordance with other analysis.

We added the stratified analysis by age groups.

3. I think the novelty of the present study is estimating the association between OL components and adiposity using fat mass. Nonetheless, comparison of the association in the present study with that in the previous studies as well as discussion of the reason for the difference or consistence are unavailable in the discussion section. This is another measure limitation of the present study. Therefore, the authors should discuss these issues.

As the studies cited on the discussion, we show the association for the obesogenic lifestyle components with the presence of overweight or obesity diagnosis by body mass index, our study adds the strength of assessing the outcome as fat mass and not only BMI. Measuring fat mass with DXA instead of using only BMI gave strength and innovation to our study, but it may be difficult to replicate given its low availability and technical difficulties involving the pediatric population making it not feasible, expensive and unpractical. To address this, we reported our results also in terms of BMI, and risk associations for OW/OB, which may be easier to transfer to the routine clinical practice.

4. Additionally, the authors should explain the underlying mechanism that derives the association between OL components and fat mass.

We add to the discussion the possible effect of the OL components in the energy imbalance. A high physical activity may prevent fat mass deposition, whereas unhealthy diet and sedentary may contribute to fat mass deposition. (Add in discussion 426-435)

5. On the other hand, the section comparing the prevalence of OB/OW as well as OL components between the present and previous studies (line 239-337) should be condensed.

We condensed these comparatives in the discussion (line 308-380)

6. The association between fat mass and BMI (line 166-175 and 191-196) may not necessary. This is because there is no rationale for investigating the association in the introduction section and the discussion about this issue is insufficient. At least, these sections should be combined and moved to the last part of the results section as section 3.4. Similarly, the results in Table 4 and Fig 1 should be shown as one table or figure, and the table (or figure) should be shown as the last table or a supplementary material.

It was important to us present the possible weakness for BMI, because could misclassify some subjects with high amount of fat mass as healthy. We change the way to present these results, moved to the last part of the section, and we eliminated table 4.

7. Introduction Please describe the reason for using fat mass as a marker of obesity additionally to BMI.

We added the information: it is important to note that we do not have reference values of BMI, and almost 12% of subjects classified as healthy weight by BMI had FMI values above the mean of OW/OB peers, which is a potential misclassification.

8. Methods Line 67-76. Please add the title to this section (e.g., 2.1. Study participants). Moreover, the authors should describe the procedure of recruitment, such as, the number of schools where participants were recruited, sampling method (i.e., random sampling or voluntary), the eligible number of participants, and the number of participants excluded from study because of missing information.

We added the information to Methods. Line 70-78.

9. Line 105-110 This section should be located after line 86.

 We made the change.

10. Line 87-93 Please detail dietary intake estimation using both 24-h recall and 12-month FFQ. Further, the authors should mention how they controlled energy misreporting because energy misreporting is common problem in self-reported dietary intake and can hamper the accurate estimation of influence of OL components on OB/OW and fat mass.

As you comment, there are some biases in the diet report. In our study we collected data through structured interviews by experienced nutritionists, 24-hour recall surveys, a local and validated food frequency questionnaire (FFQ). The 24-hour recall survey was analyzed in The Food Processor Nutrition Analysis Software® version 11.1 and it was used to estimate daily intakes of sodium and fiber. The FFQ was used to estimate intakes of fruits and vegetables, fish, and sweetened beverages. Although we found these instruments as useful and practical, they gave us important discrepancies. The 24-hour recall survey consistently reported a higher energy intake of subjects. We add it to the limitations of the study.

11. Line 113. The authors measured screen time not sitting or lying time. Additionally, screen time is considered to be more obesogenic because of the influence of advertisements of energy-dense food items. Therefore, “sedentary” should be revised as “screen time”.

We made the change in the entire document.

12. Line 118-127. The description about statistical analysis is insufficient to understand that which groups were compared and what covariates were adjusted in each analysis. Please revise this section.

We added more detail about the statistical analysis. (Line 156-186)

13. Moreover, please state the software used for statistical analysis and the alpha level for significance.

We added the information (Line 185-186).

14. Results. The number and % of participants should be shown in each sex and age group because the analysis was conducted by sex and age groups. Further, “CI95%” must be “95%CI.”

 We added number and percentage of subjects, and changed “95%CI”.

15. Line 129-133. Please include this section in the section 3.1 The response rate in line 133 should be located after the first sentence. The prevalence of overweight and obesity (line 131-133) should be shown in the section 3.2 because these variables were the outcome rather than characteristics. Thus, the values of BMI and its classification in Table 1 should be moved to Table 3.

We made the changes of the sentences and information on tables 1 and 3.

16. Line 135-160 Line 135-138 should be removed because the same information is shown in following parts (line 143-190). Moreover, line 138-140 should be moved to the last part of this section. This information should be shown after the description about each component of OL. Please revise “food intake analysis” in line 140 as “energy and macronutrient intakes” and “Calories” in Table 1 as “Energy“. Further, please add the mean of AHA Healthy Diet Score to Table 1 in accordance with other components and show Table 2 as a supplementary material.

We removed the lines 135-138, and changes 138-140, and improved the names of the variable, as you suggest. We add the diet score to table 1, and changed table 2 as supplementary material.

17. Line 162-164Please clarify the statistical method used to compare fat mass between sexes in the method section.

The information was added to statistical analysis (line 163-165).

18. Line 176-179 and 183-190 Please include these sections in the section 3.3.

The changed was made, section 3.4, line 283-300.

19. Discussion. Line 227-229 These sentences should be moved to the introduction section.

We moved the sentences (line 38-40).

20. Line 344-348. These sentences should be moved to conclusion.

We moved the sentences (line 474-480).

21. Line 349-351. There are more limitations should be mentioned, such as, generalizability of the present results, the validity of questions used to collect information on OL components, and the influence of energy misreporting on the present results.

We added more information about the limitations of the study (line 437-470).

22. Tables Please remove vertical line in Tables 2, 4, and 5, and add the number of participants to Tables 2-5. Further, please explain the statistical method used in the footnotes and add P-values to Tables.

We made the changes to the tables 2, 3, 4 and 5.

Reviewer 2 Report

The present paper aims to describe the relationships between obesogenic lifestyle-components and adiposity in children/adolescents from Mexico City.

Inactivity, sedentarism, and short-sleep were positively correlated with FMI and increased risk of Overweight and obesity. A cumulative obesogenic lifestyle (OL) score showed a significant dose-response effect with FMI. Authors concluded that the prevalence of OL-components was extremely high and associated with increased adiposity in urban Mexican children/adolescents.

The study is interesting and addressed a very important topic. However, the major concern I have is related to discussion, a limitation of the study need to be included with systematic discussion of weakness of the study.

Introduction: can be shortened

Methods: good

Results. good

Discussion: can be improved. A limitation of the study section needs to be included (see above)

Comparison with ISCOLE study and IDEFICS study is good.

A comment on differences between women and men would improve the paper (see DOI: 10.1177/2047487318810560)

Author Response

Thank you for your valuable observations and comments; they would be useful to improve the manuscript.

Below you could find your comments followed by our answers in blue. Hopefully you will be satisfied with the changes.

The study is interesting and addressed a very important topic. However, the major concern I have is related to discussion, a limitation of the study need to be included with systematic discussion of weakness of the study.

We added more information about the limitations of the study in the discussion section (section 5, line 437 – 470).

Introduction: can be shortened.

We shorted the introduction.

Discussion: can be improved. A limitation of the study section needs to be included (see above).

We added detailed information about the limitations of the study.

Comparison with ISCOLE study and IDEFICS study is good.

A comment on differences between women and men would improve the paper (see DOI: 10.1177/2047487318810560)

We added it to the discussion, in line 353-354.
